# Parvalbumin interneurons provide spillover to newborn and mature dentate granule cells

Ryan J Vaden[1†], Jose Carlos Gonzalez[1†], Ming-Chi Tsai[2], Anastasia J Niver[1], Allison R Fusilier[1], Chelsea M Griffith[1], Richard H Kramer[2], Jacques I Wadiche[1*], Linda Overstreet-Wadiche[1*]

[1]Department of Neurobiology, University of Alabama at Birmingham, Birmingham, United States; [2]Department of Molecular and Cellular Biology, University of California Berkeley, Berkeley, United States

**Abstract** Parvalbumin-expressing interneurons (PVs) in the dentate gyrus provide activity-dependent regulation of adult neurogenesis as well as maintain inhibitory control of mature neurons. In mature neurons, PVs evoke $GABA_A$ postsynaptic currents (GPSCs) with fast rise and decay phases that allow precise control of spike timing, yet synaptic currents with fast kinetics do not appear in adult-born neurons until several weeks after cell birth. Here we used mouse hippocampal slices to address how PVs signal to newborn neurons prior to the appearance of fast GPSCs. Whereas PV-evoked currents in mature neurons exhibit hallmark fast rise and decay phases, newborn neurons display slow GPSCs with characteristics of spillover signaling. We also unmasked slow spillover currents in mature neurons in the absence of fast GPSCs. Our results suggest that PVs mediate slow spillover signaling in addition to conventional fast synaptic signaling, and that spillover transmission mediates activity-dependent regulation of early events in adult neurogenesis.

**\*For correspondence:**
jwadiche@uab.edu (JIW);
lwadiche@uab.edu (LO-W)

[†]These authors contributed equally to this work

## Introduction

The dentate gyrus contains neural stem cells that continually generate new glutamatergic neurons, the granule cells (GCs), throughout life. Adult neurogenesis proceeds in a step-wise manner, with activated stem cells producing rapidly dividing progenitor cells that exit cell cycle and differentiate into newborn GCs or undergo programmed cell death, followed by a prolonged period of synaptic integration (*Toni and Schinder, 2015*). Stem cell quiescence, progenitor proliferation, survival and maturation are activity-dependent processes that enable neural network activity to influence the development of new circuitry. It is well established that GABA receptor-mediated mechanisms underlie many aspects of activity-dependent regulation, largely via $GABA_A$ receptor-mediated depolarization of proliferating progenitors and newborn neurons (*Tozuka et al., 2005*; *Ge et al., 2006*; *Jagasia et al., 2009*; *Dieni et al., 2012*; *Chancey et al., 2013*). Parvalbumin-expressing interneurons (PVs) are particularly important mediators of activity-dependent regulation, since optogenetic manipulation of PVs alters stem cell quiescence and progenitor proliferation, as well as newborn GC survival and maturation (*Song et al., 2012*; *Song et al., 2013*; *Alvarez et al., 2016*). Yet it is unclear how PVs signal to adult-born GCs at early stages of maturation.

PVs comprise just 10–20% of dentate GABAergic interneurons but have extensive axonal arbors that form powerful perisomatic synapses comprised of many release sites (*Kraushaar and Jonas, 2000*; *Hu et al., 2014*). At individual release sites, the cleft [GABA] is high and brief such that receptor kinetics primarily determine the time course of miniature $GABA_A$-receptor mediated postsynaptic currents (GPSCs) (*Overstreet et al., 2002*; *Mozrzymas, 2004*). Synchronous vesicle release across multiple active sites in combination with rapid diffusion of GABA away from spatially-segregated

sites generates large unitary GPSCs with hallmark fast rise and decay kinetics (*Bartos et al., 2002*; *Overstreet and Westbrook, 2003*; *Hefft and Jonas, 2005*; *Strüber et al., 2015*). Together with intrinsic features that enable PVs to rapidly translate synaptic excitation into action potential firing, fast IPSCs allow PVs to precisely control the time window of spike initiation and synchrony of downstream targets (*Pouille and Scanziani, 2001*; *Hu et al., 2014*; *Strüber et al., 2015*). In the dentate gyrus, PVs exhibit enriched connectivity motifs supporting lateral inhibition between GCs (*Espinoza et al., 2018*), but a relative lack of fast inhibition contributes to enhanced synaptic activation of immature GCs compared to mature GCs (*Marín-Burgin et al., 2012*; *Dieni et al., 2013*).

In contrast to mature GCs, young adult-born GCs exclusively exhibit small GPSCs with slow rise times and decay phases (*Espósito et al., 2005*; *Overstreet Wadiche et al., 2005*; *Markwardt et al., 2009*). Electrical and PV-evoked GPSCs with fast rise times do not appear until adult-born neurons are about 1 month postmitotic (*Marín-Burgin et al., 2012*; *Dieni et al., 2013*; *Alvarez et al., 2016*; *Groisman et al., 2020*). While small GPSC amplitudes are readily explained by few release sites across the small somato-dendritic domain of developing GCs, the slow kinetics are more difficult to understand. In light of the important role of PVs in regulating early events in the neurogenic cascade, we sought to understand the mechanisms underlying slow GABA$_A$ receptor signaling from PVs to newborn GCs. One possibility is that PVs innervate newly-generated GCs, but new synapses undergo a maturational process prior to supporting fast transmission (*Song et al., 2013*; *Groisman et al., 2020*). Alternatively, activation of PVs with optogenetics could recruit interneuron subtypes, such as Neurogliaform/Ivy cells, that signal via a slow form of volume transmission with a spatial-temporal [GABA] profile that is similar to spillover (*Markwardt et al., 2009*; *Karayannis et al., 2010*; *Markwardt et al., 2011*; *Overstreet-Wadiche and McBain, 2015*). However, our results exclude these to suggest that GABAergic transmission from PVs can occur via spillover to both newborn and mature GCs.

## Results

### PVs evoke slow GPSCs in newborn GCs

To examine GABAergic signaling from PVs in the DG, we generated triple-transgenic mice expressing Pomc-EGFP to identify newborn GCs, along with Pvalb^Cre-driven ChR2 (H134R)-EYFP to optogenetically activate PVs (*Figure 1—figure supplement 1A*). We first verified ChR2 activation in response to varying light pulse durations. In the presence of NBQX and CPP, we measured reliable action potentials in PVs with correlated GPSCs in mature GCs using light pulses between 0.1 and 1 ms (470 nm, *Figure 1—figure supplement 1B*). Light pulses longer than 0.5 ms typically triggered two action potentials in PVs, which in turn triggered a 2nd small peak on the decay phase of the initial GPSC, likely resulting from strong paired-pulse depression of GABA release from the same axons (*Figure 1—figure supplement 1B,C*). Cell-attached recordings revealed PV spikes reliably followed trains of light pulses at 100 Hz with at least one spike (*Figure 1—figure supplement 1D*). Light-evoked GPSCs in mature GCs were blocked by TTX (*Figure 1—figure supplement 1E*) or gabazine (n = 7, *Figure 1A*). Together these results demonstrate optogenetic activation provides sufficient depolarization for action potential-dependent GABAergic synaptic transmission from PVs.

To compare the properties of PV-mediated transmission, we made simultaneous recordings from newborn and mature GCs. Low frequency stimulation (0.1 Hz) using 1 ms light pulses generated large GPSCs in mature GCs and small but reliable GPSCs in newborn GCs, with both displaying paired-pulse depression (PPD; 200 ms ISI; *Figure 1A*). Despite a greater than 10-fold difference in GPSC amplitude (1855 ± 312 pA in mature compared to 110 ± 25 pA in newborn GCs; n = 11), there was a positive correlation between the GPSC amplitude across pairs suggesting the amplitude was dependent on the number of activated PVs (*Figure 1B*). Normalizing mature and newborn GPSCs to the peak amplitude revealed highly contrasting kinetics. Whereas PV-evoked GPSCs in mature GCs had sub-millisecond rise-times, GPSCs in newborn GCs had slower rise and decay times, and stronger PPD (*Figure 1C*). There was no correlation between GPSC amplitude and rise or decay times across cell pairs, suggesting that distinct response kinetics were independent of the number of stimulated PVs (*Figure 1D*). Light pulses < 1 ms generated very small or absent GPSCs in newborn GCs and sometimes failures in mature GCs, but the kinetic differences between newborn and mature GPSCs persisted (*Figure 1—figure supplement 2*). Together these results show that PVs generate

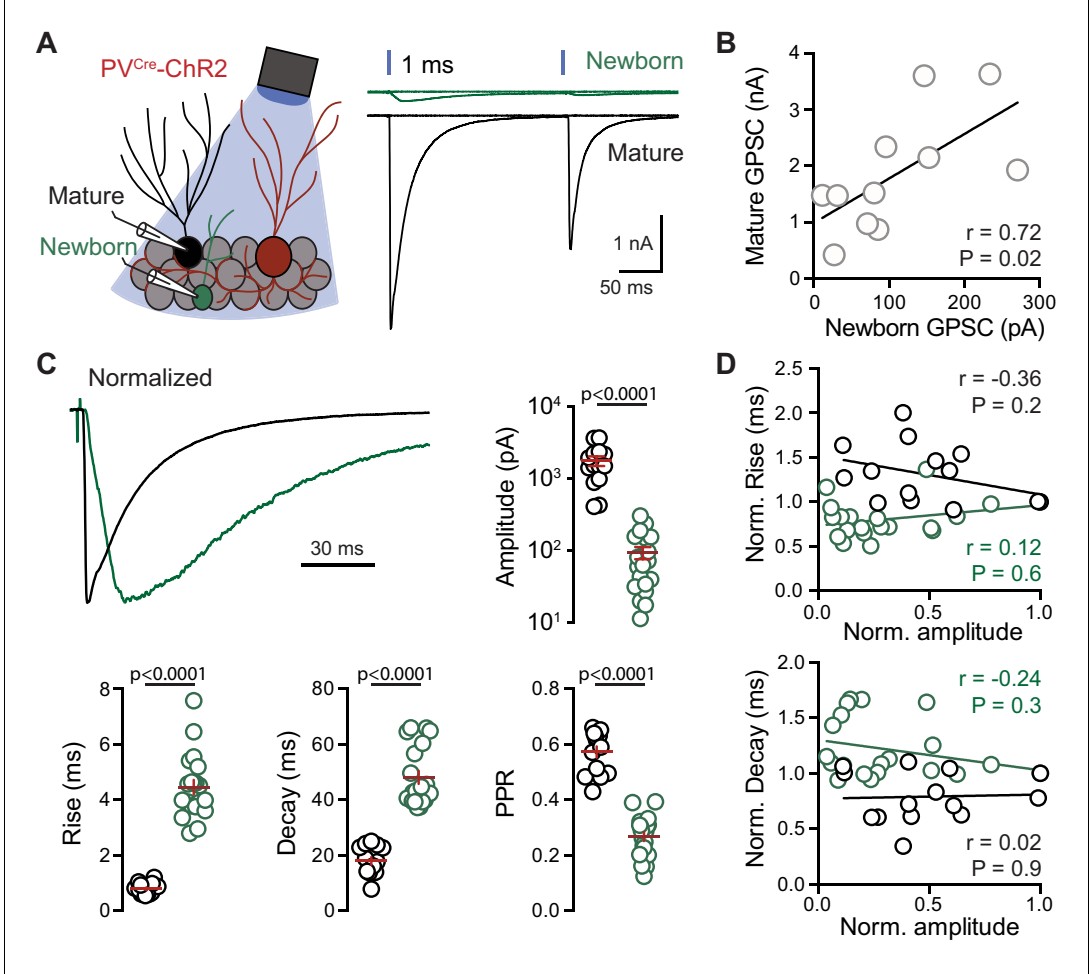

**Figure 1.** PVs generate distinct GPSCs in mature and newborn GCs. (**A**) Left, recording configuration. Right, light-evoked (200 ms ISI) GPSCs in newborn (green) and mature (black) GCs in response to 1 ms light pulseswere blocked by gabazine (10 μM). (**B**) Correlation of the average peak GPSC amplitude from dual recordings, n = 11. Spearman rank correlation. (**C**) GPSCs in mature (black) and newborn (green) GCs normalized to the peak amplitude highlights the difference in kinetics. Summary of amplitude, 20–80% rise time, weight decay τ, and PPR includes data from simultaneous (n = 11) and single cell recordings (n = 3 mature and 9 newborn GCs). Unpaired t-tests. Ampl = 1744 ± 267 pA mature, 93 ± 18 pA newborn; rise = 0.79 ± 0.05 ms mature, 4.5 ± 0.3 ms newborn; decay = 18 ± 1.4 ms mature, 48.1 ± 2.4 ms newborn; PPR = 0.57 ± 0.02 mature, 0.27 ± 0.016 newborn. (**D**) No correlation between GPSC amplitude and rise or decay time across cell pairs. n = 11.

The online version of this article includes the following figure supplement(s) for figure 1:

**Figure supplement 1.** Reliable recruitment of PVs by brief light pulses.

**Figure supplement 2.** Kinetic differences persist with sub-millisecond light pulses.

GPSCs in mature GCs with the expected fast kinetics of synapses from PV-expressing fast-spiking basket cells (*Bartos et al., 2002*), but that newborn GCs have slower GPSCs.

## Differential expression of α1 subunit cannot account for slow GPSCs

The slow GPSCs in newborn GCs could result from the expression of distinct GABA$_A$ receptors, since newborn GCs lack α1 subunits that are associated with rapid channel gating and fast decay kinetics (*Overstreet Wadiche et al., 2005*; *Karten et al., 2006*; *Picton and Fisher, 2007*; *Eyre et al., 2012*; *Dixon et al., 2014*). To address whether slow GPSCs in young GCs simply reflect the lack of α1 at immature synapses, we used genetically encoded light-regulated GABA$_A$ receptors (LiGABARs) to assess the contribution of the α1 subunit to GPSCs. In this approach, a knockin mouse with a photo-switchable α1 GABA$_A$ receptor subunit replaces its wild-type counterpart, allowing selective light-induced antagonism of α1-containing receptors (*Lin et al., 2015*). In darkness, the tethered

antagonist (PAG1C) blocks α1 containing receptors, and preconditioning with 390 nm light alleviates antagonism whereas 480 nm light reinstates it. To first test the somato-dendritic distribution of α1 subunits as observed in other brain regions (*Kerti-Szigeti et al., 2014*; *Lin et al., 2015*), we compared the light-induced block of GPSCs evoked by focal stimulation of the inner molecular layer (IML) and granule cell layer (GCL) (*Figure 2A*). In mature GCs, IML stimulation generated GPSCs with relatively slow rise and decay phases that were minimally blocked by light inactivation of α1 subunits whereas GCL stimulation evoked GPSC with faster rise and decay phases that were blocked to a greater extent by light (*Figure 2A,B*). This supports the idea that fast GPSCs arise at perisomatic synapses with high α1 content, and indeed there was a correlation between the rise time of GPSCs and the percentage of photosensitive current (*Figure 2C*). However, the time courses of fast and slow GPSCs were not altered (*Figure 2D*), illustrating that antagonizing GABA$_A$ receptors containing α1 subunits does not alter kinetics of GPSCs evoked by electrical stimulation of many synapses. While subunit-specific receptor gating dominates the time course at individual synapses (*Eyre et al., 2012*), these results are consistent with compound or multi-synaptic postsynaptic currents where additional factors like asynchrony of release and the time course of transmitter exposure contribute to rise and decay kinetics (*Diamond and Jahr, 1995*; *Mozrzymas, 2004*; *Overstreet-Wadiche and McBain, 2015*).

We then crossed α1 LiGABAR mice with GAD67-GFP reporter mice that identify adult-born GCs that are ~2–4 weeks postmitotic, thus slightly 'older' than Pomc-EGPF newborn GCs (*Cabezas et al., 2013*). We placed a focal stimulating electrode in the GCL and then recorded GPSCs evoked by the same stimulus sequentially from GFP-labeled young GCs and neighboring mature GCs (*Figure 2E*). Blocking α1 receptors with light had a smaller effect on the amplitude of GPSCs in young GCs than in mature GCs, with a measurable effect suggesting low levels of functional α1 expression (*Figure 2F*). Light-induced block of α1 subunits did not affect GPSC kinetics in either mature or immature GCs, noting again that the GPSCs in young GCs had slower rise times and decay phases compared to neighboring mature GCs (*Figure 2G*; rise time: 4.0 ± 0.4 ms vs. 1.2 ± 0.2 ms, p<0.0001; decay: 55.2 ± 3.7 ms vs. 26.2 ± 3.2 ms, p<0.0001, paired t-tests, n = 12 each). To further confirm that factors other than subunit composition contribute to the difference in GPSCs between young and mature GCs, we compared the kinetics of the isolated α1 receptor-mediated GPSCs. If receptor composition dominants, we would expect that α1-mediated GPSCs have the same kinetics in young and mature GCs. However, we observed that α1-mediated GPSCs in young GCs had both slower rise times and decay times than in neighboring mature GCs (*Figure 2—figure supplement 1*). Together these results confirm that immature GCs express lower levels of α1-containing receptors, but indicate this differential expression pattern cannot explain the slow GPSC kinetics.

## Slow GPSCs in newborn GCs have characteristics of spillover transmission

Slow GPSCs could result from a different [GABA] profile compared to the GABA transient that generates fast GPSCs. In fact, electrical stimulation generates slow GPSCs in newborn GCs with characteristics that are similar to volume transmission from neurogliaform interneurons wherein the spatial-temporal [GABA] profile dictates the slow GPSC properties rather than subunit composition (*Szabadics et al., 2007*; *Markwardt et al., 2009*; *Karayannis et al., 2010*). To compare the [GABA] underlying PV-evoked responses in newborn and mature GCs, we tested the sensitivity of light-evoked GPSCs to high and low affinity GABA$_A$ receptor competitive antagonists. Whereas high-affinity antagonists are expected to reduce synaptic currents to a similar extent regardless of the synaptic [GABA], low-affinity antagonists provide greater block of low [GABA]-mediated responses due to the fast off-rate that allows displacement of the antagonist by endogenous neurotransmitter (*Clements et al., 1992*; *Overstreet et al., 2002*; *Mozrzymas, 2004*). In this experiment, we compared sub-saturating doses of the low affinity antagonist TPMPA (200 μM) and the high affinity antagonist gabazine (GBZ, 80 nM) in the same cells to control for factors not associated with the [GABA]. We used 1 ms light pulses to enhance the signal-to-noise ratio of newborn GPSCs. As expected GBZ, blocked the peak amplitude of mature and newborn GPSCs to the same extent (51 ± 9% verses 51 ± 4%, paired t-test, p=0.9, n = 4). However, the low-affinity antagonist TPMPA (200 μM) blocked GPSCs in newborn GCs to a greater extent than GPSCs in mature GCs (*Figure 3A*; paired t-test, p<0.001, n = 4). This differential effect suggests that PV-evoked GPSCs in newborn GCs are mediated by a lower [GABA] than in mature GCs.

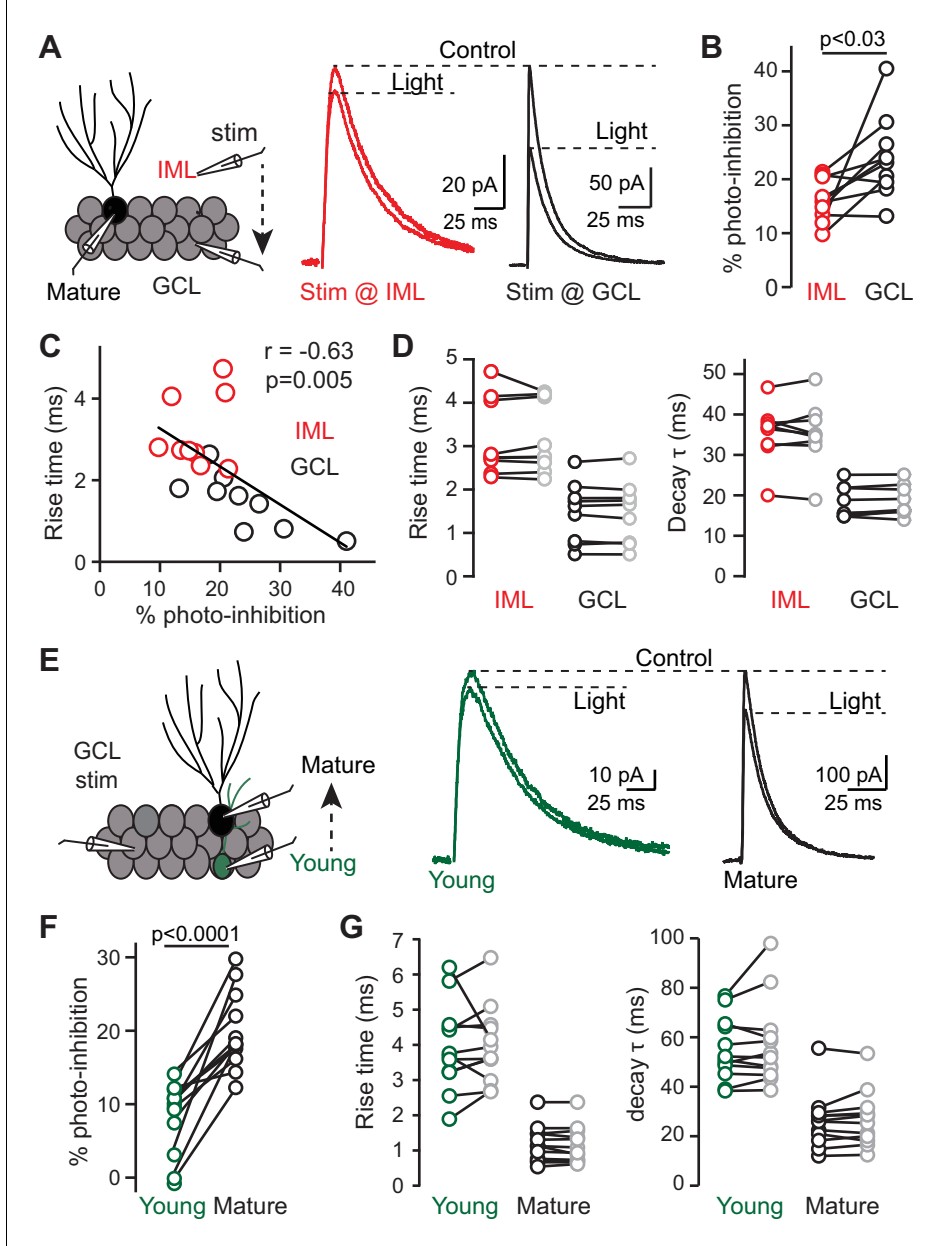

**Figure 2.** Differential expression of α1 subunits does not account for slow PSCs. (**A**) Left, recording from mature GCs with sequential stimulation in the IML (red) and GCL (black). Right, GPSCs showed a smaller α1-GABA$_A$R contribution when stimulating the IML (red) compared to GCL (black). Stimulus artifacts blanked. (**B**) Photoinhibition reduced IML GPSCs by 16 ± 1% whereas GCL GPSCs were reduced by 24 ± 3%. Paired t-test, n = 9. (**C**) Slow rise times correlated with less photoinhibition, suggesting a lower contribution of synaptic α1-GABA$_A$Rs (n = 18). (**D**) Photoinhibition did not change the rise or decay times. Rise time with IML stimulation: ctrl, 4.7 ± 0.4 ms; blocking α1-GABA$_A$R, 4.8 ± 0.4 ms; with GCL stimulation: ctrl, 2.0 ± 0.3 ms; blocking α1-GABA$_A$R, 2.0 ± 0.3 ms; Decay time with IML stimulation: ctrl, 37.1 ± 2.4 ms; blocking α1-GABA$_A$R, 37.3 ± 2.5 ms; with GCL stimulation: ctrl, 21.9 ± 2.4 ms; blocking α1-GABA$_A$R, 23.4 ± 2.3 ms; n = 9. (**E**) Left, stimulation in the GCL while sequentially recording from young and mature GCs in G42:α1-LiGABAR mice. Right, slow GPSCs in young GCs (green) showed less photoinhibition compared to mature GCs (black). (**F**) Photoinhibition reduced GPSCs in young GCs by 8 ± 1% whereas mature GPSCs were reduced by 20 ± 2%; n = 12 each, paired t-test. (**G**) Photoinhibition did not change the rise or decay time of GPSCs. Rise time: young, ctrl, 4.0 ± 0.4 ms; blocking α1-GABA$_A$R, 4.0 ± 0.3 ms; mature, ctrl, 1.2 ± 0.2 ms; blocking α1-GABA$_A$R, 1.2 ± 0.2 ms; Decay time: young control, 55.2 ± 3.7 ms; blocking α1-GABA$_A$R, 57.4 ± 5.0 ms; mature: control, 26.2 ± 3.2 ms; blocking α1-GABA$_A$R, 27.0 ± 3.2 ms. The online version of this article includes the following figure supplement(s) for figure 2:

*Figure 2 continued on next page*

*Figure 2 continued*

**Figure supplement 1.** α1-mediated GPSCs in young and mature GCs.

---

Neurotransmitter that escapes the synaptic cleft can reach receptors at a distance from the release site to generate spillover transmission. In addition to slow rise and decay phases, a hallmark characteristic of spillover transmission is high sensitivity to transporters that terminate the action of fast neurotransmitters outside the synapse (*Isaacson et al., 1993*; *Coddington et al., 2014*). We found that PV-evoked GPSCs in mature and newborn GCs were differentially affected by the GAT1 inhibitor NO711 (2 µM; *Figure 3B*). In newborn GCs, NO711 robustly increased GPSC amplitude (from 82 ± 15 pA to 193 ± 40 pA), and prolonged both the 20–80% rise time (from 4.6 ± 0.6 ms to 13 ± 1.7 ms) and weighted decay (from 52 ± 2.6 ms to 136 ± 12.3 ms; paired t-tests, n = 9). In mature GCs, NO711 slightly *decreased* the amplitude of the GPSC (from 1980 ± 244 pA to 1476 ± 187 pA) with no effect on the rise time (from 0.77 ± 0.08 ms to 0.84 ± 0.07 ms), and NO711 prolonged the weighted decay phase (from 18 ± 1 ms to 69 ± 5 ms; paired t-tests, n = 11). The amplitude reduction

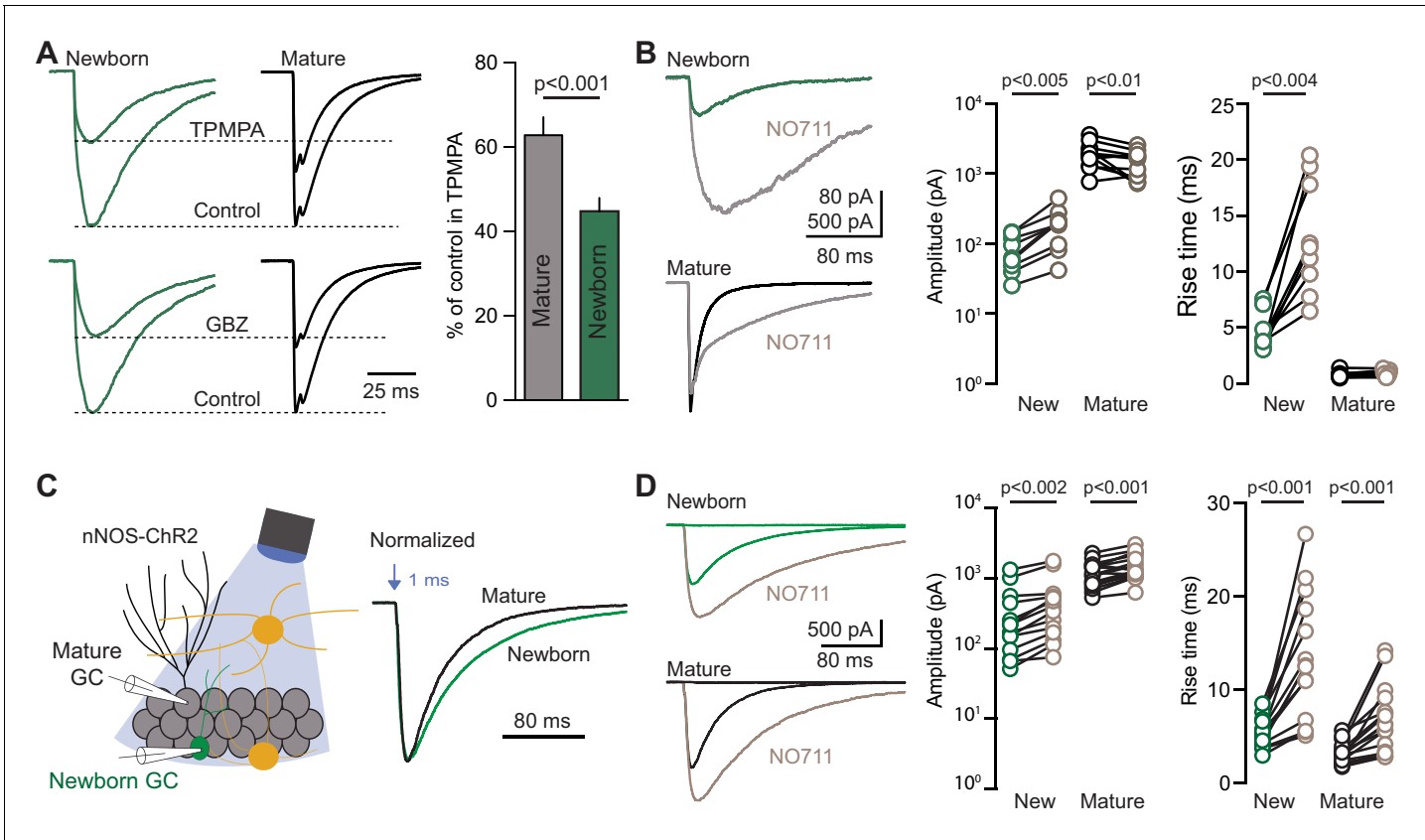

**Figure 3.** Slow GPSCs in newborn GCs have characteristics of spillover. (**A**) Representative PV-evoked GPSCs in simultaneous recordings of newborn (green) and mature GCs (black) in TPMPA (200 µM; top) and GBZ (80 nM; bottom). TPMPA reduced GPSCs in mature GCs to 63 ± 4% of control and in newborn GCs to 45 ± 3% of control, n = 4, p<0.001, paired t-test. (**B**) In newborn GCs, NO711 (2 µM) increased the GPSC amplitude and prolonged both the rise time and weighted decay τ (from 52 ± 2.6 ms to 136 ± 12.3 ms; n = 9, paired t-tests). In mature GCs, NO711 decreased GPSC amplitude (1980 ± 244 pA to 1476 ± 187 pA) and prolonged the decay τ (18 ± 1 ms to 69 ± 5 ms) but did not alter the rise time of GPSCs (n = 11; paired t-tests). (**C**) Recording configuration to identify GPSCs evoked by nNOS-expressing interneurons in Pomc-EGFP expressing newborn (green) or mature (black) GCs. Representative GPSCs evoked by 1 ms light pulse normalized to peak amplitude highlight slow rise and decay phases. (**D**) In contrast to GPSCs evoked by PVs (**B**). NO711 (gray) increased the amplitude and rise time of nNOS-evoked GPSCs in both newborn (n = 14) and mature GCs (n = 18, paired t-tests). Recordings from newborn and mature GCs were interleaved in separate slices.

The online version of this article includes the following figure supplement(s) for figure 3:

**Figure supplement 1.** Reliable recruitment of nNOS interneurons by brief light pulses.

in mature GCs could result from either activation of presynaptic $GABA_B$ receptors by increased ambient GABA or postsynaptic $GABA_A$ receptor desensitization (*Overstreet et al., 2000*; *Overstreet and Westbrook, 2001*). Importantly, the increase in the amplitude and rise time of newborn GPSCs by NO711 supports the idea that synaptic currents are mediated by GABA acting outside of the synaptic cleft.

The slow kinetics, high sensitivity to TPMPA and robust effects of NO711 of PV-evoked GPSCs in newborn GCs are characteristic of GPSCs evoked by ivy/neurogliaform interneurons, slow-spiking GABAergic interneurons that signal via volume transmission that lacks postsynaptic anatomical specializations (*Szabadics et al., 2007*; *Oláh et al., 2009*; *Karayannis et al., 2010*; *Overstreet-Wadiche and McBain, 2015*). A large fraction of ivy/neurogliaform cells express neuronal nitric oxide synthase (nNOS) (*Tricoire et al., 2010*; *Gonzalez et al., 2018*; *Christenson Wick et al., 2019*), so to compare slow GPSCs evoked by PVs and ivy/neurogliaform interneurons, we also bred nNOS-CreER:(H134R)-EYFP:Pomc-EGFP mice that were treated with tamoxifen after weaning (*Figure 3C*). In contrast to PVs, nNOS interneurons exhibited extensive processes in the hilus and molecular layer but not the GCL, and light-pulses up 5 ms in duration triggered single rather than multiple spikes (*Figure 3—figure supplement 1A,B*). Brief light pulses generated slow $GABA_B$-GIRK IPSCs in mature GCs (*Gonzalez et al., 2018*) as well as slow $GABA_A$ IPSCs blocked by gabazine (*Figure 3—figure supplement 1C*). As expected for transmission from neurogliaform interneurons, nNOS-evoked GPSCs in mature GCs had slower rise and decay times compared to PV-evoked GPSCs (*Figure 3—figure supplement 1D*). Comparison of nNOS-evoked GPSCs (1 ms light pulses, in the $GABA_B$ antagonist CGP55845) revealed exclusively slow synaptic responses in both newborn and mature GCs (*Figure 3C*). GPSCs were about 4-fold smaller in newborn GCs (344 ± 104 pA, n = 14 versus 1221 ± 113 pA, n = 18, p<0.0001) and had slower rise (5.6 ± 0.5 ms versus 3.2 ± 0.3 ms, p=0.003) and decay times (87 ± 6 ms versus 53 ± 5 ms, p=0.0001, unpaired t-tests). Importantly, in contrast to PV-evoked GPSCs, NO711 (5 μM) increased the amplitude and rise time of GPSCs in both newborn and mature GCs (*Figure 3D*). These results show that optogenetic stimulation of nNOS-expressing interneurons generate GPSCs consistent with volume transmission from neurogliaform interneurons, and that slow GPSCs in newborn GCs from both PV and nNOS interneurons are generated by a spatial-temporal [GABA] profile that differs from typical mature PV synapses. Interestingly, nNOS-evoked GPSCs in newborn GCs exhibited larger amplitudes and slower decay times than PV-evoked slow GPSCs (*Figure 3—figure supplement 1E*), suggesting volume transmission might provide more robust signaling than PV-mediated spillover.

## PV-ChR2 targets fast-spiking basket cells

A small fraction of PVs is reported to co-express nNOS (*Jinno and Kosaka, 2002*; *Shen et al., 2019*), raising the possibility that slow GPSCs in newborn GCs elicited by PV-ChR2 actually arise from neurogliaform interneurons. We thus sought to identify the interneuron subtypes targeted by PV-Cre and compare results to interneurons targeted by nNOS-CreER. First, we assessed PV-ChR2-YFP co-labeling with PV and found a high degree of co-localization, with 84% of YFP-ChR2[+] cells co-localized with PV (262 cells from 3 mice) and 63% of PV[+] cells were co-localized with YFP-ChR2 (351 cells from 3 mice; *Figure 4A*). We wondered whether unreliable detection of somata by membrane-targeted ChR2/YFP affected these measures (*Figure 4—figure supplement 1*), so to facilitate visualization of soma in both fixed and acute slices, we also used offspring from Cre mouse lines crossed with Ai14 (tdTomato; tdT) reporter mice. This approach yielded similar results, with 77% of PV-tdT expressing interneurons showing strong PV immunoreactivity (102/133 PV-tdT cells) and 56% of PV immunoreactive cells co-expressing PV-tdT (102/183, n = 2 mice; *Figure 4—figure supplement 2A*). These results suggest high specificity but moderate efficiency of this PV-Cre line, noting that low levels of PV are likely below our threshold for detection. Importantly, there were <2% of PV-tdT cells with evidence of nNOS co-immunoreactivity (3/168 PV-tdT cells) and <1% of nNOS immunoreactive cells expressing PV-tdT (3/405 cells, n = 2 mice; *Figure 4—figure supplement 2B*). We also assessed PV and nNOS expression in slices from nNOS-CreER/tdT mice and found that 77.7% of nNOS-tdT labeled cells expressed nNOS but only 0.3% expressed PV (746 cells from 3 mice, not shown). Thus, we found little evidence for significant overlap between interneurons targeted by PV-Cre and nNOS-CreER.

Second, we made whole cell recordings with post-hoc reconstructions from PV interneurons expressing PV-tdT (*Figure 4B*) or PV-ChR2 and compared results to recordings from labeled

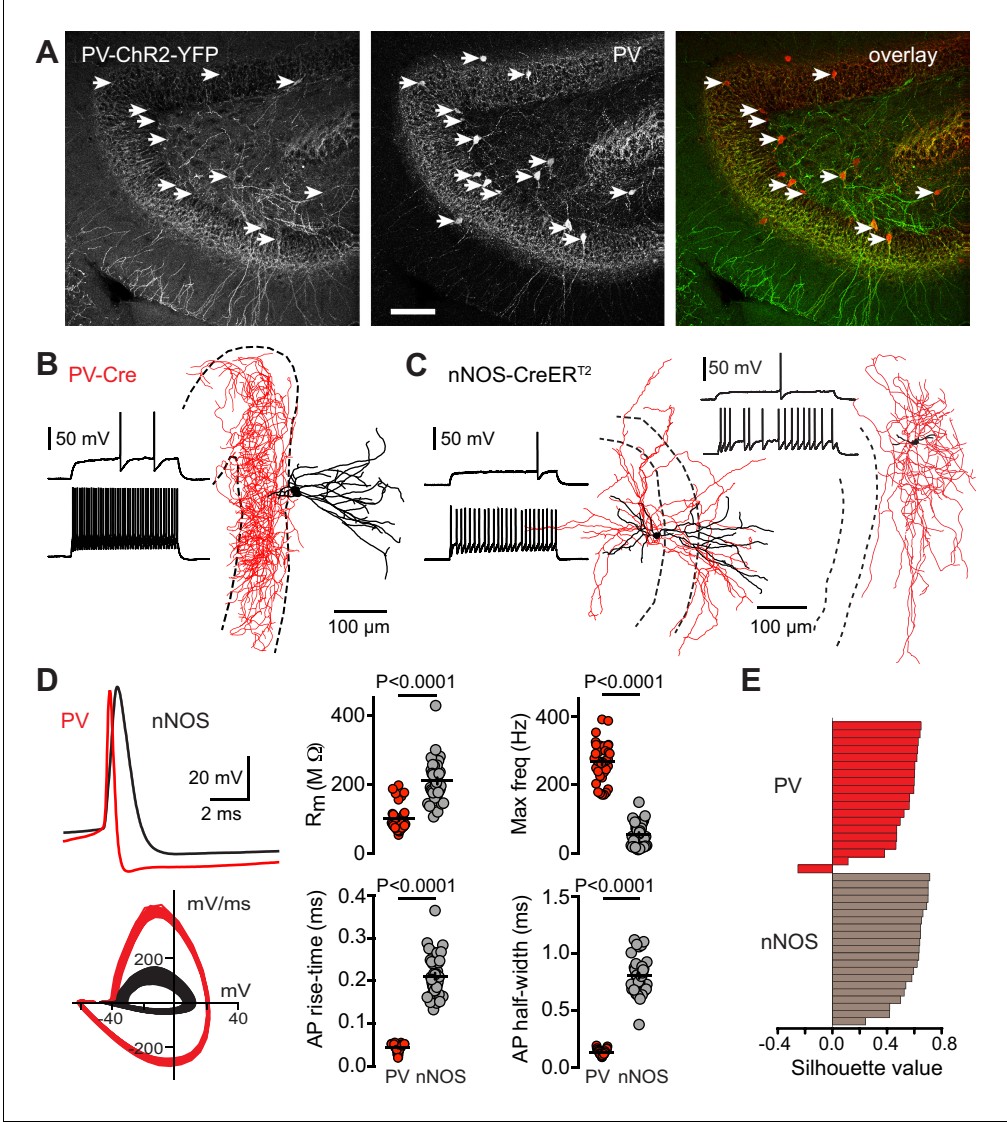

**Figure 4.** PV-Cre targets fast-spiking basket cells. (**A**) *Left*, confocal z-projection (16 images at 2 μm steps, 20X, 0.5 NA) showing PV-ChR2-YFP expression and immunolabeling of PV (*middle*). Arrows indicate labeled soma. *Right*, overlay showing high co-localization between ChR2-YFP and PV, with arrows indicating co-localized soma. Scale bar, 100 μm. (**B**) Biocytin reconstruction of a PV showing typical basket cell morphology with axon (red) targeting the granule cell layer (dotted lines). Inset shows spiking pattern at threshold and 2x threshold current injections. (**C**) Partial reconstructions of two nNOS interneurons, with spiking patterns as in (**B**). (**D**) Top, PVs showed rapid action potential kinetics (red) compared to nNOS interneurons (black). Bottom, overlaid phase plots of voltage responses at 2x threshold. Right, summary of action potential properties of PVs (n = 38) and nNOS interneurons (n = 39). Max frequency was measured in response to a step current injection of 700 pA. (**E**) Silhouette plot from a 2-step cluster analysis of 15 parameters that resulted in two clusters comprised of nNOS and PVs (see *Figure 4—figure supplement 3*). Within each cluster, cells are ranked on the vertical axis in decreasing order of silhouette value (n = 40 cells).

The online version of this article includes the following figure supplement(s) for figure 4:

**Figure supplement 1.** PV-ChR2 co-labeling with PV immunoreactivity.
**Figure supplement 2.** PV-tdT with PV and nNOS immunoreactivity.
**Figure supplement 3.** PVs have basket cell morphology.
**Figure supplement 4.** Comparison of cluster analysis parameters.

interneurons in nNOS-CreER mice (*Figure 4C*). PV interneurons exhibited low input resistance (103 ± 6 MOhm), fast action potential kinetics (rise time: 0.044 ± 0.001 ms; half-width: 0.14 ± 0.004 ms) and high spiking rates (268 Hz ±9 Hz; n = 38) consistent with expected properties of fast-spiking basket cells or axo-axonic cells (*Figure 4D*). All reconstructed PVs had basket or axo-axonic morphology with axons targeting the GCL and dendrites in the hilus and ML (12/12, *Figure 4—figure supplement 3*). In contrast, nNOS interneurons exhibited higher input resistance (209 ± 9 MOhm), slower action potential kinetics (rise time: 0.21 ± 0.008 ms; half-width: 0.81 ± 0.02) and low maximal firing rates (56 ± 6 Hz; n = 39). None of the reconstructed nNOS interneurons exhibited basket cell morphology (*Figure 4B*). A two-step unbiased cluster analysis using 15 parameters revealed that PV and nNOS expressing interneurons are well separated populations with the highest silhouette value of 0.6 corresponding to two clusters (*Figure 4E*; *Figure 4—figure supplement 4*). Together these results confirm that PV-Cre is primarily expressed by fast-spiking basket cells and axo-axonal cells whereas nNOS-Cre targets slow-spiking interneurons including neurogliaform and long-range interneurons (*Gonzalez et al., 2018*; *Christenson Wick et al., 2019*).

Despite the minimal apparent overlap between PV and nNOS-expressing interneurons targeted by our Cre lines, residual co-labeling could be functionally relevant since activation of single neurogliaform cells is sufficient to evoke slow GPSCs in newborn GCs (*Markwardt et al., 2011*). Furthermore, neurogliaform cells exhibit gap junction coupling with PVs (*Simon et al., 2005*), raising the intriguing possibility that optogenetic activation of PVs could lead to indirect recruitment via gap-junction mediated depolarization (*Apostolides and Trussell, 2013*). In fact, we found that in the presence of GABA receptor blockers, long light pulses could generate current shifts in slow-spiking (non-ChR2 expressing) interneurons that were sensitive to the gap-junction blocker carbenoxolone (*Figure 5—figure supplement 1*). To further test the possibility that sparse or indirect recruitment of neurogliaform cells contributes to PV-ChR2-evoked slow GPSCs in newborn GCs, we probed functional characteristics of GABAergic transmission that are known to differentiate somatic-projecting fast-spiking PVs and dendritic-projecting neurogliaform interneurons.

First, we assessed presynaptic regulation of GABA release. Unlike for PVs, GABA release from neurogliaform cells generates homosynaptic suppression by presynaptic GABA$_B$ receptors that enhance PPD (*Poncer et al., 2000*; *Price et al., 2008*). However, we found that the GABA$_B$ receptor antagonist CGP55845 (2 μM) had no effect on the amplitude or PPR of light-evoked GPSCs in newborn GCs from PV-ChR2 slices. In contrast, CGP55845 increased the PPR of GPSCs evoked in mature GCs from nNOS-ChR2 slices (*Figure 5A*). This differential regulation of presynaptic release properties suggests that PV-evoked GPSCs in newborn GCs do not arise from neurogliaform cells.

Second, we assessed postsynaptic GABA$_B$-mediated signaling because the ability to generate GABA$_B$ receptor-mediated slow IPSCs also differentiates synaptic transmission from PVs and neurogliaform cells (*Tamás et al., 2003*; *Price et al., 2008*; *Gonzalez et al., 2018*). We previously reported that single light pulses (1 ms) generates robust GABA$_B$ receptor-mediated IPSCs in mature GCs from nNOS-ChR2 mice (21 ± 3 pA, n = 7), but not in GCs from PV-ChR2 mice (0.8 pA ±0.2 pA, n = 4; unpaired t-test, p=0.001; *Gonzalez et al., 2018*). We also assessed GABA$_B$ IPSCs generated by shorter light pulses. First, we confirmed interneuron recruitment by 0.1 ms pulses in each slice by GABA$_A$ GPSCs measured at a holding potential 20 mV above E$_{Cl}$ (*Figure 5B*, top). At E$_{Cl}$, PVs failed to generate GABA$_B$-receptor mediated slow PSCs whereas slow outward PSCs that were potentiated by NO711 and blocked by CGP55845 were readily apparent under the same conditions in nNOS-ChR2 slices (*Figure 5B*, bottom). Together, these results show that both pre- and postsynaptic properties of PV-evoked GPSCs are inconsistent with non-specific or indirect recruitment of neurogliaform cells.

## Spillover GPSCs are generated cooperatively by multiple PVs

Spillover requires cooperative nonlinear pooling of transmitter from multiple release sites, thus it is more sensitive to reducing the density of active release sites compared to the originating direct synaptic responses (*Scanziani, 2000*; *Overstreet and Westbrook, 2003*). To confirm the distinct modes of transmission to mature and newborn GCs from PVs, we compared the sensitivity of GPSCs to a stimulus protocol designed to reduce the density of active release sites and thus the pooling between sites. We used a conditioning train stimulus (4 repetitions of 20 Hz light pulses) to transiently deplete GABA release, using interleaved control and preconditioning episodes (*Figure 6—figure supplement 1A*). The preconditioning stimuli effectively reduced GABA release (which

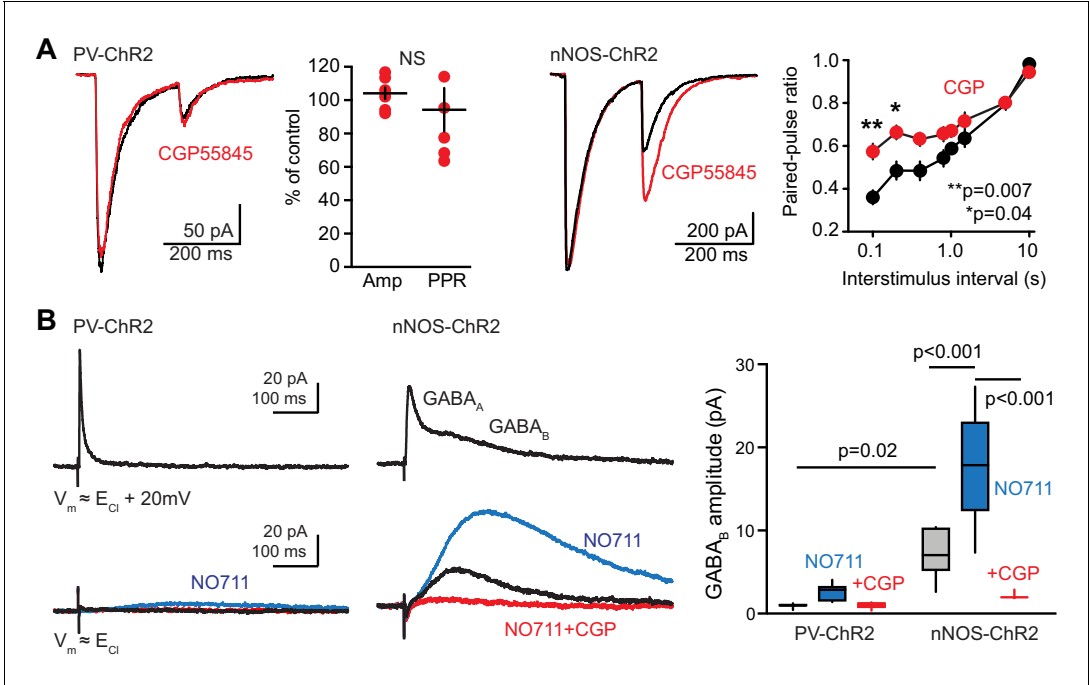

**Figure 5.** Characteristics of GABA release are consistent with PVs. (**A**) Left, the GABA_B receptor antagonist CGP55845 (2 μM, red) had no effect on the amplitude or paired-pulse ratio (PPR) of PV-evoked GPSCs in newborn GCs. Paired t-tests, n = 6, p=0.5 for amplitude and p=0.2 for PPR. In contrast, CGP55845 increased the PPR of GPSCs in slices from nNOS-ChR2 mice at inter-stimulus intervals of 100 ms (**p=0.007) and 200 ms (*p=0.04). Paired t-tests with multiple comparison corrections, n = 7. (**B**) Mature GCs were held at 20 mV above experimentally-identified $E_{Cl}$ to first confirm that brief light pulses (0.1 ms) generated GABA release assayed by GABA_A PSCs (top traces). PV-mediated GABA_B GPSCs were undetectable in control conditions. In slices from nNOS-ChR2 mice, GABA_B IPSCs were apparent at both $E_{Cl}$ and $E_{Cl}$+ 20 mV, enhanced by NO711 and blocked by CGP55845 (lower traces). Right, summary data from both mouse lines. ANOVA with Tukey multiple comparisons, n = 4–7 cells.

The online version of this article includes the following figure supplement(s) for figure 5:

**Figure supplement 1.** PV-ChR2 photo-responses in non-PV interneurons.

recovered after a 30 s interval, not shown), and increased the PPR (*Figure 6—figure supplement 1B*) consistent with vesicle depletion and thus a reduction in the density of active release sites (*Foster and Regehr, 2004*; *Vaden et al., 2019*). Importantly, the conditioning stimulus reduced GPSCs in newborn GCs to a greater extent than GPSCs in mature GCs (78 ± 2% versus 54 ± 5%, *Figure 6A*), consistent with a high sensitivity to cooperative pooling of the slow GPSC.

To further investigate how PVs generate spillover to newborn GCs, we performed simultaneous recordings between PVs and newborn GCs. With current-clamp recording from a PV, we used increasing duration light pulses to assess the threshold of PV spiking while also monitoring GABA release to the voltage-clamped newborn GC (*Figure 6B*, left). With subthreshold light stimulation for the recorded PV, a small GPSC in the newborn GC was initially evident, and it increased in amplitude as additional PV(s) were recruited at increasing light durations (0.15 ms to 1 ms). In *Figure 6B*, right, the recorded PV was recruited at a 0.2 ms light duration, revealing that multiple PVs typically contribute to spillover in newborn GCs. As expected from the low unitary connectivity reported previously from a large number of recordings (*Markwardt et al., 2011*), 3/5 PVs failed to evoke a unitary GPSC. But in 2/5 recordings, a unitary GPSC was generated in response to current injection to the PV (*Figure 6C*; average amp = 7.8 pA, rise = 2.8 ms, n = 2). As expected for pooling from multiple release sites, the decay of GPSCs evoked by single PVs had faster kinetics (average decay τ = 28 ms) than GPSCs evoked by light stimulation (*Figure 6C*, inset). Together these results show that single PVs can generate GPSCs in newborn GCs, but suggest that optogenetic stimulation generates GABA pooling from many active PVs.

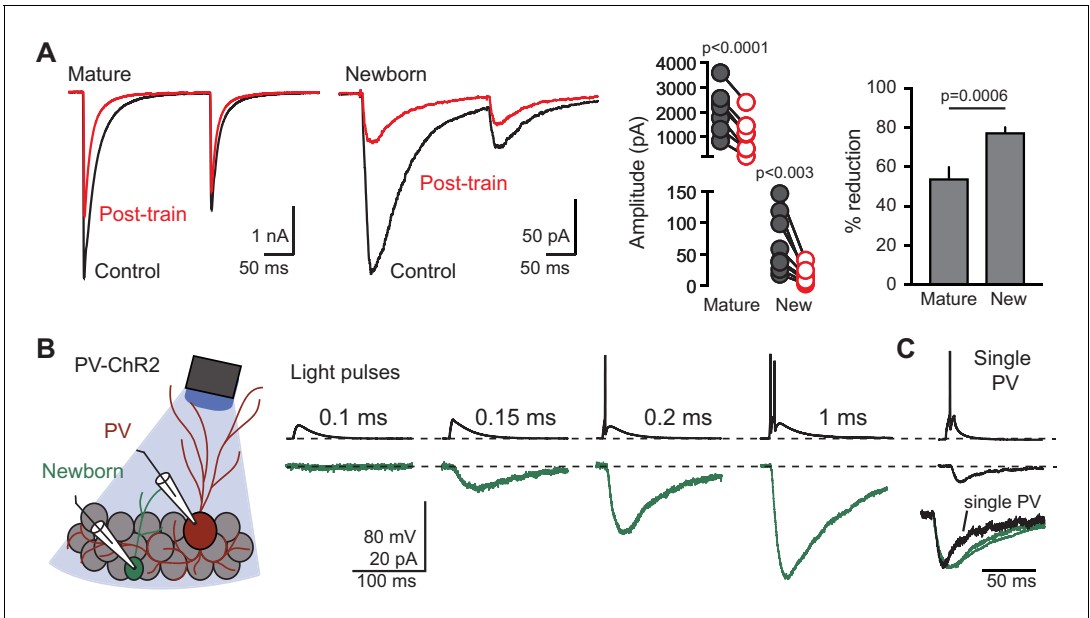

**Figure 6.** Spillover GPSCs are generated cooperatively by multiple PVs. (**A**) Left, examples GPSCs in mature (black) and newborn GCs (green) in response to a conditioning train that depletes GABA release. Middle and right, the conditioning train reduces GPSCs (paired t-tests) to a greater extent in newborn GCs (78 ± 2%, n = 8) compared to mature GCs (54 ± 5%, unpaired t-test, n = 7). (**B**) Cartoon and examples of simultaneous recordings from a PV (red) and newborn GC (green) in response to increasing durations of photostimulation at low light intensity. The threshold for this PV was 0.2 ms light duration. Single (top) and average of 3–6 traces (bottom) are shown. (**C**) In the same pair of cells, direct current injection to the PV (10 ms, 700 pA) generated a small GPSC in the newborn GC, showing that single PVs can generate spillover to newborn GCs (found in 2/5 experiments). Inset, normalized GPSCs in response to single PV stimulation and optogenetic (multiple) PV stimulation shows slower decay phase, as expected for GABA pooling in response to multiple active PVs.

The online version of this article includes the following figure supplement(s) for figure 6:

**Figure supplement 1.** Conditioning stimulation reduces GABA release.

## PVs generate spillover to mature GCs

Our results are consistent with PVs signaling to newborn GCs via spillover, a mechanism that does not require anatomically-defined synaptic specializations (*Szapiro and Barbour, 2007*; *Coddington et al., 2013*). But an alternative explanation is that PV innervation arises at functionally immature synapses, since close appositions between PV axon terminals and progenitors have been described using immunoelectron microscopy (*Song et al., 2013*). To test whether PV-evoked GPSCs represent functionally immature synapses targeted to newborn GCs or non-specific spillover signaling, we assessed whether similar slow GPSCs could be evoked in mature GCs.

Spillover due to pooling between neighboring release sites can prolong the decay component of fast GPSCs even following GABA release from a single interneuron (*Overstreet and Westbrook, 2003*; *Biró and Nusser, 2005*). Indeed, a slow component of the GPSC in mature GCs was often evident following optogenetic stimulation and this component was enhanced by NO711 (*Figure 3*). To investigate the subcellular localization of this slow component, we compared the reversal potential of the PV and nNOS-evoked GPSCs. The peak of the PV-GPSC reversed near the calculated $E_{Cl^-}$ whereas the nNOS-GPSC reversed at a more hyperpolarized potential, as expected for somatic and dendritic localized synapses, respectively (*Groisman et al., 2020*). Importantly, the slow component of the PV-GPSC (measured 50 ms after the stimulus) had a similar reversal potential as the peak, consistent with a somatic localization (*Figure 7—figure supplement 1*). Both reducing the duration of the light pulse or using the precondition stimulus to reduce the density of active sites speeded the GPSC decay phase in mature GCs, consistent with spillover contributing the slow decay phase (*Figure 7—figure supplement 2A,B*). Even the reduction of GABA release during paired-pulse stimuli generated a detectable speeding of 2nd GPSC in both mature GCs (half-width reduced to 8.9 ± 0.6 ms from 11.8 ± 1.0 ms, n = 30, p<0.001 paired t-test) and newborn GCs (reduced to 31.4 ± 1.8 ms

from 41.9 ± 1.7 ms, n = 21, p<0.001 paired t-test). Prolongation of the mature GPSC decay phase by NO711 also depended on the density of active release sites (*Figure 7—figure supplement 1C*). However, both reducing the light duration and the precondition train tended to reduce the number of PV spikes per stimulus, potentially contributing to the speeding of the decay. Thus, we also tested whether mature GCs slow GABA GPSCs from PVs in the absence of the fast GPSC, as occurs in newborn GCs.

To isolate a spillover component, we took advantage of distinct thresholds for optogenetic recruitment to activate PVs that are not synaptically connected by reducing the light duration and intensity to induce failures of fast GPSCs. In an example of a simultaneous recording from a mature and newborn GC, reducing the light intensity initially produced smaller GPSCs with variable latencies and very fast rise times in mature GCs, presumably reflecting GABA release from single axons with different thresholds of activation (*Figure 7A*, blue symbols and traces). At lowest light intensities, fast GPSCs often failed but close inspection of the apparent failures (red traces) revealed both failure or small GPSCs with slow rise times (*Figure 7A* right). Averaging 'successes' in mature GCs when fast GPSCs failed revealed a slow GPSC (black trace, *Figure 7B*), that was larger than the correlated averaged GPSC in the newborn GC, perhaps due to the high number of failures (green trace; *Figure 7B*). Comparing only successes revealed that slow GPSCs in the mature (black) and newborn (green) GC had similar amplitudes and rise times that were dramatically different from fast GPSCs (blue; *Figure 7B*). The decay time course of the slow GPSC in mature GCs was slightly faster than the decay phase of GPSCs in newborn GCs (normalized traces, *Figure 7B*, right). The isolated slow GPSC in mature GCs also scaled to the slow decay phase of the fast GPSC (dotted line, *Figure 7B* right), suggesting that this spillover component accounts for the slow decay phase in fast GPSCs that is enhanced by NO711 (*Figure 3B*). By sorting traces in mature GCs at just-threshold light intensities for generating fast GPSCs (see methods), we found isolated slow GPSCs in 7/12 mature GCs with an amplitude of 19 ± 2 pA and rise times of 2.9 ± 0.5 ms (n = 7). Consistent with spillover signaling, NO711 (2 μM) increased the amplitude and rise time of these slow GPSCs (*Figure 7C*). In contrast, in mature GCs that did not exhibit slow GPSCs at sub-threshold light intensities, NO711 had no effect on the fast GPSCs generated at threshold light intensities (*Figure 7D*, n = 3). Thus, these independent synaptic and spillover GPSCs in mature GCs are similar to fast and slow GPSCs in cerebellar GCs that arise from directly-connected and remote terminals, respectively (*Rossi and Hamann, 1998*). These results suggest that PVs can signal to both mature and newborn GCs via spillover transmission, whereas only mature GCs exhibit direct synaptic connections.

## Discussion

Here we addressed how PVs signal to adult-born GCs to regulate early stages of adult neurogenesis prior to establishment of functionally mature synapses. PV-evoked GABAergic currents in newborn GCs were small in amplitude with slow kinetics, and highly sensitive to a low affinity GABA$_A$ receptor antagonist and blockade of GABA transport. These observations raised the possibility of non-specific or indirect recruitment of neurogliaform interneuron-mediated volume transmission which exhibits similar properties. However, characterization of nNOS co-labeling and PV physiology made this possibility unlikely, and PV-evoked GPSCs failed to exhibit pre- and postsynaptic hallmarks of neurogliaform-mediated transmission. Conversely, reducing the number of active PV release sites affected the amplitude of GPSCs in newborn GCs to a greater extent than in mature GCs, pointing to the importance of extrasynaptic transmitter pooling in generating slow responses. By activating PVs that did not directly innervate recorded mature GC, we also unmasked PV-mediated slow GPSCs in the absence of hallmark fast GPSCs. Together these results demonstrate two distinct spatiotemporal profiles of PV-mediated neurotransmission; whereas mature GCs exhibit fast GPSCs expected at PV synapses, both mature and newborn GCs can receive slow GPSCs that result from spillover.

### Mechanisms underlying slow GPSCs from PVs

Studies investigating PV-mediated regulation of early stages in adult neurogenesis have consistently reported small GPSCs with slow rise and decay phases in newborn progeny, with fast GPSCs appearing only several weeks after cell birth (*Song et al., 2013*; *Alvarez et al., 2016*; *Groisman et al., 2020*). Our results show that these early synaptic currents display properties of spillover, a mode of signaling that does not require anatomically-defined synapses but is facilitated by a high density of

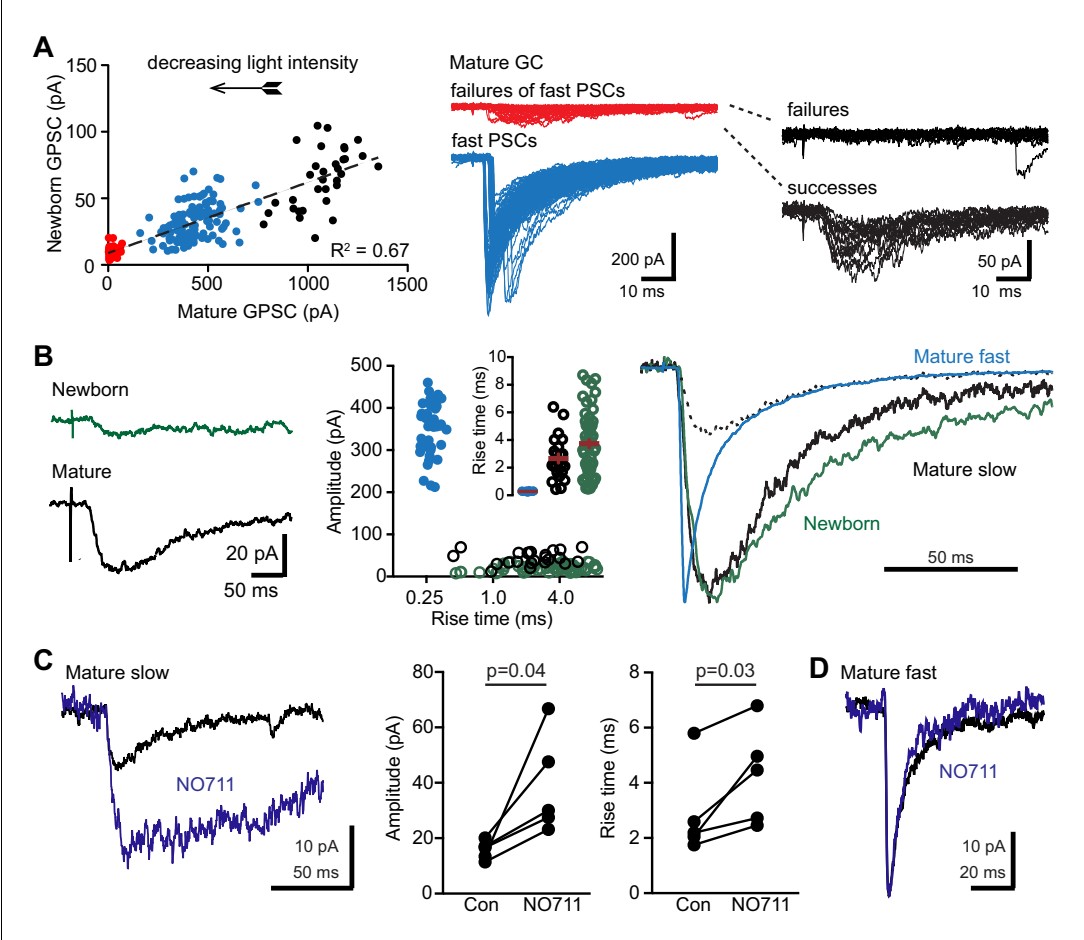

**Figure 7.** PVs evoke spillover in mature GCs. (**A**) Reducing the light intensity (arrow; 0.1 ms duration) decreased the amplitude of GPSCs in mature and newborn GCs. At intermediate intensities, GPSCs in mature GCs had fast rise times (blue symbols and traces; 0.2 ± 0.005 ms) and variable latencies, illustrating variability in threshold for activating axons. Below threshold for fast GPSCs, small slow events were evident (red symbols and traces) that were sorted into failures and successes (right, black traces). (**B**) Left, average of the subthreshold successes in the mature GC (black) and the average of the correlated traces in the newborn GC (green, successes and failures combined). Middle, comparison of amplitudes and rise times of fast GPSCs in mature (blue), slow GPSCs in mature (black) and slow GPSCs in newborn GCs (green). Right, normalized slow GPSCs in mature and newborn GCs have similar rise and decay phases. Dotted line shows slow GPSC in mature GC scaled to the decay of the fast GPSC. (**C**) NO711 increased the amplitude and rise time of slow GPSCs in mature GCs, consistent with spillover. Paired t-tests, n = 5. In contrast, in mature GCs that did not exhibit slow GPSCs near threshold light stimulation, NO711 did not affect fast GPSCs (**D**).

The online version of this article includes the following figure supplement(s) for figure 7:

**Figure supplement 1.** Reversal potentials of fast and slow GPSCs.

**Figure supplement 2.** Reducing the number of active synapses limits spillover in mature GCs.

release sites where transmitter pooling between sites activates extrasynaptic or neighboring synaptic receptors (*Kullmann, 2000*). Spillover-mediated signaling is prominent at specialized synapses in the cerebellum, where it can occur between neighboring cells even in the absence of anatomically-defined synaptic contacts (*Rossi and Hamann, 1998*; *Szapiro and Barbour, 2007*; *Coddington et al., 2013*; *Coddington et al., 2014*; *Nietz et al., 2017*). Since anatomically-defined synaptic specializations and close appositions have been reported between PVs and retroviral-labeled DG progenitors, an alternative possibility is that slow GPSCs are generated at functionally immature synapses (*Song et al., 2013*). We favor spillover for three reasons. First, the ability to isolate spillover currents in mature GCs indicates slow currents are not exclusive to newborn GCs. Second, PV-mediated GPSCs are slower than GPSCs at immature PV synapses during the first postnatal week that exhibit rise times of <1 ms and decay phases of <10 ms (*Doischer et al., 2008*). Finally, while maturation of presynaptic properties could contribute to changes in PPD of PV-evoked GPSCs

(*Groisman et al., 2020*), an alternative explanation is that desensitization of postsynaptic receptors accounts for some portion of PPD in newborn GCs (*Figure 1C*; *Karayannis et al., 2010*). Thus, a parsimonious conclusion is that slow GPSCs result from GABA spillover rather than immature synapses targeted to newborn GCs. However, spillover-like transmission to newborn and mature GCs could also arise from transient or orphan presynaptic boutons undergoing structural plasticity (*Tyagarajan and Fritschy, 2014*; *Wierenga, 2017*).

There is no doubt that GABA$_A$ receptor subunit composition contributes to GPSC properties, with the α1 subunit specifically associated with the maturation of fast GPSC decay kinetics (*Pelkey et al., 2017*). However, we found that GPSCs mediated by α1-containing receptors had slow kinetics in newborn GCs, confirming that factors other than subunit composition dictate the GPSC time course (*Figure 2—figure supplement 1*). While subunit composition is the main factor determining the the decay phase of synaptic currents at an individual synapse (i.e. mIPSCs *Eyre et al., 2012*), additional factors come into play when tens to hundreds of release sites are activated together. For example, asynchrony of transmitter release is the major determinant of the rise and decay time of evoked EPSCs whereas channel gating accounts for the rise and decay of mEPSCs generated at single sites (*Diamond and Jahr, 1995*). Asynchronous release could explain why manipulating α1-containing GABA$_A$ receptors had little effect on the kinetics of the fast rise and decay components of PV-evoked GPSCs. Spillover between release sites, limited by GABA transporters, could explain the slow decay component (*Overstreet and Westbrook, 2003*). Our results highlight that the relative contribution of receptor composition, release asynchrony and the spatial-temporal profile of the [GABA] transient depends on the types of synaptic responses being studied.

Our data are consistent with young GCs expressing low levels of functional α1-containing receptors and mature GCs showing preferential localization of α1 receptors at somatic synapses. While our experiments do not address the subunit composition underlying slow GPSCs, based on early expression in newborn GCs and contribution to slow PSCs in other brain regions, it is likely to include α5 containing receptors (*Overstreet Wadiche et al., 2005*; *Zarnowska et al., 2009*; *Deprez et al., 2016*). While α1-containing synapses can exhibit faster mIPSC deactivation kinetics than non-α1 containing synapses (*Eyre et al., 2012*), detection of subunit-dependent gating requires that receptors are exposed to similar [GABA] transients. At conventional synapses, single vesicles generate a [GABA] profile at postsynaptic receptors that peaks in the low mM range and decays within a ms (*Overstreet et al., 2002*; *Mozrzymas, 2004*; *Barberis et al., 2011*). In contrast, volume transmission from neurogliaform interneurons (see below) exposes receptors to a [GABA] that peaks below 100 μM and decays over tens of milliseconds (*Karayannis et al., 2010*). Based on the similar GPSC rise times, as well as TPMPA and NO711 sensitivity, the [GABA] profile resulting from PV spillover is likely similar to volume transmission from neurogliaform cells, with slower decay times in newborn GCs from both interneuron subtypes potentially reflecting subunit composition or a lack of postsynaptic receptor clustering (*Figures 3C* and *7B*). Thus, the different properties of PV-evoked GPSCs in newborn and mature GCs primarily reflects the distinct spatiotemporal [GABA] profile with differences in receptor subunit composition or distribution playing only a supporting role.

## Slow GPSCs from multiple interneuron subtypes

Our experiments show that newborn GCs exhibit slow GPSCs evoked by both PV and nNOS interneurons. In mature neurons, the kinetics of GPSCs from neurogliaform interneurons depends on the spatial-temporal profile of [GABA], and not receptor subunit composition or release synchrony (*Szabadics et al., 2007*; *Karayannis et al., 2010*). This is consistent with GABA being released into the extrasynaptic space rather than at conventional anatomical synapses (*Oláh et al., 2009*; *Overstreet-Wadiche and McBain, 2015*). Thus, unlike PVs, neurogliaform interneurons also recruit extrasynaptic GABA$_B$ receptor activation following single stimuli (*Scanziani, 2000*; *Tamás et al., 2003*; *Gonzalez et al., 2018*). We previously described electrically-evoked and unitary volume transmission to Pomc-EGFP newborn GCs (*Markwardt et al., 2009*; *Markwardt et al., 2011*) and now confirm robust optogenetic-evoked volume transmission from nNOS-expressing neurogliaform interneurons to both mature and newborn GCs (*Figure 3C,D*, new *Figure 3—figure supplement 1*). Volume transmission represents the normal mode of transmission for neurogliaform interneurons and thus GPSCs in newborn and mature GCs have similar characteristics. In contrast, PV signaling to newborn and mature GCs has distinct characteristics that result from GABA released at mature GC synapses 'spilling over' to act on newborn neurons. Spillover results from transmitter acting beyond the

primary target of receptors clustered within the synaptic cleft and thus can be considered a form of non-specific signaling. We speculate that the reliance of spillover signaling on cooperativity from multiple PVs (*Figure 6*) makes it particularly suitable for dynamically regulating experience-dependent neurogenesis, whereas volume transmission from nNOS interneurons might provide more consistent phasic depolarization, in accordance with the distinct behaviorally-state dependent firing properties of these interneuron classes (*Lapray et al., 2012*).

## Functional consequences of slow GPSCs

Slow GPSCs regulate activity-dependent maturation and survival of newborn GCs via depolarization (*Ge et al., 2006*; *Jagasia et al., 2009*; *Song et al., 2013*; *Alvarez et al., 2016*). Slow GPSCs are first detected in a minority of GC progenitors (*Wang et al., 2005*), arising from PV but not somatostatin or vasoactive intestinal peptide-expressing interneurons (*Song et al., 2013*). Pomc-EGFP identifies an early postmitotic neuronal stage when essentially all newborn GCs exhibit slow GPSCs and first acquire functional excitatory synapses from hilar mossy cells (*Overstreet et al., 2004*; *Chancey et al., 2014*). GABAergic depolarization allows voltage-dependent unblock of NMDARs that is required for activity-dependent incorporation of AMPARs at functionally silent NMDAR-only synapses (*Chancey et al., 2013*). Both glutamatergic mossy cells and mature GCs recruit disynaptic GABA release to newborn GCs, with the slow time course of the GABAergic conductance matching the time course of the NMDAR-EPSC, suggesting circuitry for coupling GABA and glutamate signaling that together allow rapid synaptic integration of newborn GCs in response to network activity (*Chancey et al., 2014*; *Alvarez et al., 2016*). Interestingly, fast GPSCs from PV and SST interneurons do not appear in adult-born GCs until >4 weeks post-mitosis, when dendrites have extended through the molecular layer and perforant path excitatory synapses are established (*Espósito et al., 2005*; *Mongiat et al., 2009*; *Dieni et al., 2013*; *Groisman et al., 2020*). The dependence on non-canonical mechanisms of GABAergic synaptic depolarization at early stages suggests that newborn GCs might lack the molecular machinery for conventional GABAergic synapse formation.

While slow $GABA_A$ signaling provides a robust source of depolarization for newborn GC maturational processes, the functional significance of slow GPSCs in mature GCs is less clear considering PVs exhibit multiple mechanisms for generating fast inhibition with high temporal precision (*Bartos et al., 2002*; *Hefft and Jonas, 2005*; *Bartos and Elgueta, 2012*; *Hu et al., 2014*). Interestingly, the duration and amplitude of unitary GPSCs differs with distance between individual PVs and postsynaptic GCs, and these differences may confer non-uniformity that enables emergence of functionally independent focal gamma bursts (*Strüber et al., 2015*; *Strüber et al., 2017*). Thus, it is plausible that slow spillover from PVs, dependent on the density of active terminals, can also contribute to non-uniformity of $GABA_A$ conductances to support focal gamma frequency oscillations.

# Materials and methods

**Key resources table**

| Reagent type (species) or resource | Designation | Source or reference | Identifiers | Additional information |
|---|---|---|---|---|
| Genetic reagent (*M. musculus*) | Pomc-EGFP | Jackson Laboratory | Stock #: 009593 RRID:MGI:3776090 | |
| Genetic reagent (*M. musculus*) | PV<sup>Cre</sup> | Jackson Laboratory | Stock #: 008069 RRID:MGI:3590684 | |
| Genetic reagent (*M. musculus*) | Ai14 | Jackson Laboratory | Stock #: 007914 RRID:MGI:3809523 | |
| Genetic reagent (*M. musculus*) | Ai32 | Jackson Laboratory | Stock #: 024109 RRID:MGI:104735 | |

*Continued on next page*

*Continued*

| Reagent type (species) or resource | Designation | Source or reference | Identifiers | Additional information |
|---|---|---|---|---|
| Genetic reagent (*M. musculus*) | nNOS-CreER | Jackson Laboratory | Stock #: 014541 RRID:MGI:97360 | |
| Genetic reagent (*M. musculus*) | α1-LiGABAR | Jackson Laboratory | Stock #: 028965 RRID:MGI:95613 | |
| Genetic reagent (*M. musculus*) | G42 | Jackson Laboratory | Stock #: 007677 RRID:MGI:3721279 | |
| Antibody | anti-GFP, Alexa Fluor 488 (rabbit polyclonal) | Invitrogen | Cat#: A-21311; RRID:AB_221477 | 1:1000 |
| Antibody | anti-nNOS (rabbit polyclonal) | Chemicon | Cat#: AB5380 | 1:2000 |
| Antibody | anti-PV (rabbit polyclonal) | Abcam | Cat#: ab11427 | 1:1000 |
| Antibody | goat anti-rabbit BIOT | Southern Biotech | Cat#: 4050–08; RRID:AB_2732896 | 1:800 |
| Antibody | streptavidin, Alexa Fluor 647 | Invitrogen | Cat#: S32357 | 1:200 |
| Chemical compound, drug | carbenoxolone (CBX) | Sigma-Aldrich | Cat#: C4790; CAS: 7421-40-1 | 100 μM |
| Chemical compound, drug | NBQX | Abcam | Cat#: ab120045; CAS: 118876-58-7 | 10 μM |
| Chemical compound, drug | CPP | Abcam | Cat#: ab120159; CAS: 126453-07-4 | 5 μM |
| Chemical compound, drug | SR95531 (gabazine) | Abcam | Cat#: ab120042; CAS: 104104-50-9 | 80 nM, 3–10 μM |
| Chemical compound, drug | NO711 | Sigma-Aldrich | Cat#:N142; CAS: 145645-62-1 | 2–5 μM |
| Chemical compound, drug | TTX | Abcam | Cat#: ab120054; CAS: 4368-28-9 | 1 μM |
| Chemical compound, drug | QX-314 | Abcam | Cat#: ab120118; CAS: 5369-03-9 | 50 μM |
| Chemical compound, drug | TPMPA | Sigma-Aldrich | Cat#: T200; CAS: 182485-36-5 | 200 μM |
| Chemical compound, drug | CGP55845 | Sigma-Aldrich | Cat#: SML0594; CAS: 149184-22-5 | 2–10 μM |
| Chemical compound, drug | PAG-1C | Synthesized in-house; *Lin et al., 2018* | | 25–50 μM |
| Chemical compound, drug | tris(2carboxyethyl) phosphine (TCEP) | Sigma-Aldrich | Cat# 646547 | 2.5–5.0 mM |

*Continued*

| Reagent type (species) or resource | Designation | Source or reference | Identifiers | Additional information |
|---|---|---|---|---|
| Software, algorithm | Axograph X, version | AxoGraph Scientific | axograph.com | |
| Software, algorithm | pClamp 10 | Molecular Devices | moleculardevices.com | |
| Software, algorithm | Prism, version 7/8 | GraphPad | graphpad.com | |
| Software, algorithm | Neurolucida | MBF Bioscience | mbfbioscience.com | |

All animal procedures followed the Guide for the Care and Use of Laboratory Animals, U.S. Public Health Service, and were approved by the University of Alabama at Birmingham Institutional Animal Care and Use Committee (IACUC-21289) and University of California, Berkeley Animal Care and Use Committee (AUP-2016-04-8700).Mice of either gender were maintained on a 12 hr light/dark cycle with ad libitum access to food and water. Mouse lines used to generate experimental mice included *Pomc*-EGFP (*Overstreet et al., 2004*), *Pvalb*[Cre] (PV[Cre]; Jax #008069), Ai14 (Jax#007914), Ai32 (Jax #024109), *Nos1*-Cre[ER] (nNOS-CreER: *Taniguchi et al., 2011*; Jax #014541), *Gad1*-GFP (G42; Jax#007677) and α1-LiGABAR (Jax#028965). nNOS-CreER mice were fed tamoxifen-containing chow (Teklad250) for one week after weaning. α1-LiGABAR mice contain a single amino acid mutation in the α1 subunit that allows conjugation of a 'photoswitch' that renders the receptors light-sensitive (*Lin et al., 2015*). α1-LiGABAR mutant (α1-T125C) homozygosity was confirmed by genotyping (*Lin et al., 2015*). We used GFP expression from the G42 line to identify young GCs (*Cabezas et al., 2013*) in G42:α1-LiGABAR mice.

## Electrophysiology

Mice were anesthetized and intracardially perfused with ice-cold cutting solution containing the following (in Mm): 110 choline chloride, 25 glucose, 7 $MgCl_2$, 2.5 KCl, 1.25 $Na_2PO_4$, 0.5 $CaCl_2$, 1.3 Na-ascorbate, 3 Na-pyruvate, and 25 $NaHCO_3$, bubbled with 95% $O_2$/5% $CO_2$. The brain was removed and 350–400 µm slices were prepared using a vibratome (Leica VT1200S). Slices were then incubated at 37°C for 30 min in recording solution containing the following (in mM): 125 NaCl, 2.5 KCl, 1.25 $Na_2PO_4$, 2 $CaCl_2$, 1 $MgCl_2$, 25 $NaHCO_3$, and 25 glucose bubbled with 95% $O_2$/5% $CO_2$ before transfer to room temperature. Patch pipettes were typically filled with the following (in mM): 140 KCl, 4 $MgCl_2$, 10 EGTA, 10 HEPES, 4 Mg-ATP, 0.3 Na-GTP, 7 Phosphocreatine (pH 7.3 and 310 mOsm). For recordings from interneurons, postsynaptic $GABA_B$ experiments and measuring $E_{GABA}$, the intracellular solution contained (in mM): 135 Kgluconate, 2 $MgCl_2$, 0.1 EGTA, 10 HEPES, 4 KCl, 2 Mg-ATP, 0.5 Na-GTP, 10 Phosphocreatine (pH 7.3, 310 mOsm, and 2–4 MΩ). Biocytin (0.2%) was included in some recordings. All experiments were performed at 32°C and at a holding potential of −80 mV unless otherwise noted. Recordings were performed in NBQX (10 µM) and CPP (5 µM). Currents were sampled at 20–40 kHz and filtered at 2-10 kHz (MultiClamp 700B; Molecular Devices) using Pclamp10 (Molecular Devices) or AxographX (AxoGraph Scientific). At least 10–20 individual GPSCs were recorded under stable conditions and averaged to calculate effects of photoactivation or drug treatments.

For recordings from α1-LiGABAR or G42:α1-LiGABAR mice, the intracellular solution contained (in mM): 145 Cs-gluconate, 0.3 EGTA, 10 HEPES, 10 CsCl, 4 Mg-ATP, 0.4 Na-GTP (pH 7.3, 310 mOsm, and 3–4 MΩ). Newborn or mature GCs were voltage-clamped at +5 mV and currents were sampled at 10 KHz. Prior to experiments, slices were treated with tris(2-carboxyethyl)phosphine (TCEP; 2.5–5 mM, 5–10 min), washed, and then treated with photoswitching compound PAG-1C (25–50 µM, 25–45 min) at room temperature to convert the α1-GABAR-T125C1 mutant receptors into LiGABARs. GPSCs were evoked by focal stimulation using glass electrode pulled from theta borosilicate glass (# 2BF150-86-10; Sutter). Photocontrol of α1-LiGABARs was achieved by illuminating the slices through 20x objective with conditioning light of either 390 nm (unblocking) or 480 nm (blocking) (Lumencor).

Activation of ChR2 used a 455 nm LED mounted in the epifluorescence light path controlled by an external driver (DC2100; ThorLabs), with duration controlled by the TTL outputs of pClamp. Unless noted, 1 ms light pulses were used. We cannot verify the duration of sub-millisecond light pulses but in every case reducing the duration from 1.0 to 0.1 ms at 0.25 intervals reduced GPSC amplitudes culminating in failures of fast GPSCs. In experiments to isolate fast and slow GPSCs in mature GCs, 100–200 episodes using light durations at threshold for evoking fast GPSCs (0.2–0.5 ms;~50% failures) were manually sorted by defining fast GPSCs as events with a rise time <1 ms. Averaging all failures of fast GPSCs revealed slow GPSCs in 7/12 mature GCs that were subsequently manually sorted for small slow events and true failures (as in *Figure 7A*).

Two step cluster analysis was performed on 40 labeled cells from the DG of PV-Cre/tdT and nNOS-CreER/tdT mice using IBM SPSS version 24 (IBM Corp., Armonk, New York, USA). Eleven continuous variables were used (input resistance, rheobase, threshold, AP amplitude, AP rise-time, AP half-width, AHP amplitude, AHP rise-time, AHP half-width, maximum firing frequency, and accommodation ratio) along with four categorical variables (molecular marker, cell body location, presence of fast notch in AHP, and firing pattern at twice threshold). Log-likelihood was used as the distance measure, the number of clusters was automatically determined, and Akaike's information criterion (AIC) was used as the clustering criterion. The robustness of clustering was quantified using the silhouette measure with the highest silhouette value of 0.6 corresponding to two clusters.

Drugs were purchased from Abcam or Sigma-Aldrich and were used at the following concentrations: NBQX (AMPAR antagonist, 10 µM), CPP (NMDAR antagonist, 5 µM), SR95531 (gabazine; $GABA_A$R antagonist 80 nM, 3–10 µM), NO711 (GAT-1 antagonist, 2–5 µM), TTX ($Na^+$-channel blocker, 1 µM), QX-314 ($Na^+$-channel blocker, 50 µM), TPMPA ($GABA_A$R low affinity antagonist, 200 µM), and CGP55845 ($GABA_B$R antagonist, 2–10 µM). Other salts and chemicals were purchased from Fischer Scientific.

## Immunohistochemistry

Anesthetized mice were intracardially perfused with 0.9% NaCl or 0.1M PBS followed by chilled 4% PFA and extracted brains were fixed overnight in 4% PFA. Free-floating horizontal section through the entire brain (50 µm; Vibratome 1000) were stored at −20°C in antifreeze (30% ethylene glycol, 20% glycerol in PBS). Slices were washed for ten minutes in TBST (0.5% Triton-X-100 in 1% TBS). To enhance endogenous YFP/GFP, slices were blocked (0.1M TBS, 1% glycine, 3% bovine serum albumin, 0.4% Triton X-100% and 10% normal goat serum) and incubated overnight with anti-GFP conjugated Alexa 488 (1:1000, Invitrogen, Carlsbad, CA). For PV and nNOS immunostaining, citrate buffer (10 mM sodium citrate with 0.5% tween 20, pH 6) was heated to boiling then applied to sections and allowed to incubate for 20 min. Slices were washed two times in TBST then incubated in 0.3% hydrogen peroxide for 15 min. Nonspecific binding was blocked by incubating slices in 10% NGS in TBST for one hour. Slices were protected from light while incubated in primary antibody diluted 1:2000 in TBST with rabbit anti-nNOS (Chemicon, ab5380), 1% NGS, and 0.1% sodium azide or rabbit anti-Parvalbumin (Abcam, ab11427) diluted 1:1000 in the same solution for 48 hr at room temp. Slices were washed three times in TBST then incubated for three hours with goat anti-rabbit biotinylated antibody (Southern Biotech, 4050–08) diluted 1:800 in TBST with 1% NGS. Slides were washed in 1x PBS then mounted with ProLong Gold (Invitrogen) and allowed to cure for 24 hr. Acute brain slices containing biocytin-filled cells were post-fixed in 4% PFA+ picric acid for at least 24 hr, then washed with TBST and incubated in fluorescent conjugated streptavidin (Alexa 647; Invitrogen, S32357) diluted 1:200 in TBST for 30 min at room temperature. Morphological reconstruction and analysis were performed with Neurolucida (MBF Bioscience).

For assessing colocalization, single sections were imaged using a 20x oil-immersion objective (0.85 NA) on an Olympus FluoView 300 confocal microscope or using a 20x water-immersion objective (0.5 NA) on an Olympus Fluoview FV1200. Using ImageJ and the cell counter plugin, cells in each channel were first identified independently, and then images were overlaid and cells that aligned in the x, y and z planes were counted as co-labeled.

## Data analysis

Data are expressed as mean ± SEM. Group comparisons used two-sample paired or unpaired t-tests or ANOVA when number of groups exceeded two. All tests used two-tailed Type I error rate of 0.05

using Prism7 or 8 (GraphPad Prism, La Jolla, CA). The p values are indicated in all figures and n values are indicated in the legends.

## Acknowledgements

We thank the members of the Wadiche labs for helpful comments throughout this project and Mary Seelig for excellent technical assistance. This work was supported by NIH F31NS098553 (RJV), NS064025 & NS105438 (LOW), NS065920 & NS113948 (JIW) and NS100911 (RHK).

## Additional information

### Competing interests

Linda Overstreet-Wadiche: Reviewing editor, *eLife*. The other authors declare that no competing interests exist.

### Funding

| Funder | Grant reference number | Author |
|---|---|---|
| National Institute of Neurological Disorders and Stroke | F31NS098553 | Ryan J Vaden |
| National Institute of Neurological Disorders and Stroke | NS064025 | Linda Overstreet-Wadiche |
| National Institute of Neurological Disorders and Stroke | NS105438 | Linda Overstreet-Wadiche |
| National Institute of Neurological Disorders and Stroke | NS065920 | Jacques I Wadiche |
| National Institute of Neurological Disorders and Stroke | NS113948 | Jacques I Wadiche |
| National Institute of Neurological Disorders and Stroke | NS100911 | Richard H Kramer |

The funders had no role in study design, data collection and interpretation, or the decision to submit the work for publication.

### Author contributions

Ryan J Vaden, Conceptualization, Formal analysis, Funding acquisition, Investigation, Visualization, Writing - original draft, Writing - review and editing; Jose Carlos Gonzalez, Conceptualization, Formal analysis, Validation, Investigation, Visualization, Writing - review and editing; Ming-Chi Tsai, Conceptualization, Resources, Formal analysis, Investigation, Methodology, Writing - review and editing; Anastasia J Niver, Formal analysis, Investigation, Writing - review and editing; Allison R Fusilier, Formal analysis, Validation, Investigation, Visualization, Writing - review and editing; Chelsea M Griffith, Formal analysis, Validation, Investigation, Writing - review and editing; Richard H Kramer, Conceptualization, Resources, Funding acquisition, Methodology, Writing - review and editing; Jacques I Wadiche, Conceptualization, Data curation, Formal analysis, Supervision, Funding acquisition, Writing - review and editing; Linda Overstreet-Wadiche, Conceptualization, Formal analysis, Supervision, Funding acquisition, Project administration, Writing - review and editing

### Author ORCIDs

Ryan J Vaden (iD) https://orcid.org/0000-0001-8544-4003
Jose Carlos Gonzalez (iD) https://orcid.org/0000-0003-4612-1943
Ming-Chi Tsai (iD) https://orcid.org/0000-0002-6216-8672
Richard H Kramer (iD) https://orcid.org/0000-0002-8755-9389
Jacques I Wadiche (iD) https://orcid.org/0000-0001-8180-2061
Linda Overstreet-Wadiche (iD) https://orcid.org/0000-0001-7367-5998

## Ethics

Animal experimentation: All animal procedures followed the Guide for the Care and Use of Laboratory Animals, U.S. Public Health Service, and were approved by the University of Alabama at Birmingham Institutional Animal Care and Use Committee (IACUC-21289) and University of California, Berkeley Animal Care and Use Committee (AUP-2016-04-622 8700).

## Decision letter and Author response

Decision letter https://doi.org/10.7554/eLife.54125.sa1
Author response https://doi.org/10.7554/eLife.54125.sa2

## Additional files

### Supplementary files

- Transparent reporting form

### Data availability

All data points analyzed during the study are illustrated in the figures.

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
