## [Decision Letter]

**Acceptance summary:**

This study provides new optogenetic evidence that in immature adult-born dentate granule cells, GABAergic signaling from parvalbumin cells is mediated entirely by spillover transmission rather than direct synaptic transmission, and requires pooling from several different PV terminals. These results provide an explanation for previously confusing observations of very slow kinetics of PV-mediated GABAergic synaptic currents in immature granule cells.

**Decision letter after peer review:**

Thank you for submitting your article "Parvalbumin interneurons provide spillover to newborn and mature dentate granule cells" for consideration by *eLife*. Your article has been reviewed by four peer reviewers, one of whom is a member of our Board of Reviewing Editors, and the evaluation has been overseen by Gary Westbrook as the Senior Editor. The following individual involved in review of your submission has agreed to reveal their identity: Eric Schnell (Reviewer #2). Of note, Gary Westbrook served only an administrative role in this manuscript, and did not participate in the reviews or the post-review discussion to avoid the appearance of COI with the senior author who was a postdoc with Westbrook many years ago.

The reviewers have discussed the reviews with one another and the Reviewing Editor has drafted this decision to help you prepare a revised submission.

Summary:

The paper provides compelling evidence that GABAergic signaling from PV cells to immature granule cells is entirely mediated by spillover transmission, and go further to demonstrate that PV-mediated spillover transmission occurs onto mature GCs as well. Although some details replicate the prior work of Markwardt et al., this study provides a convincing and detailed explanation for the slow kinetics of PV-mediated GABAergic responses in immature adult-born cells, the mechanisms underlying PV-signaling to adult-born neurons, and may even upend the view that adult-born granule cells receive direct synaptic innervation by interneurons prior to excitatory synaptogenesis. As such, it is important progress towards a full understanding of the circuit integration of adult-born cells and the control of their maturation.

Essential revisions:

1) Co-localization for ChR2/PV should be done in the PV-ChR2 mice using immunostaining.

2) Please show that the slow GPSCs in adult are blocked by a GABAA receptor antagonist.

3) Paired PV-immature GC recordings (currently n=5 with 2 that showed spillover) could offer some additional insights into spillover transmission, namely in terms of whether it is somewhat cell-specific vs. "volume" transmission, etc.

4) One surprising result in Figure 2G is that full blockade of the α1 subunit responsible for the fast kinetics produces no change in the rise time in new or mature GCs, in contrast to what would have been expected, an increase in rise time. This inconsistency must be addressed.

5) IPSC decay was faster after pre-conditioning pulses, presumably due to less GABA release. By this argument, the second IPSC during paired-pulse experiments in which there is paired-pulse depression should also be faster – is this the case? This should be addressed convincingly, using either new experiments (TPMPA) or with reanalysis of existing data if they are convincing. E.g., one might expect that the second response (in immature neurons) would be *more* sensitive to TPMPA than the first one.

---

## [Author Response]

Essential revisions:1) Co-localization for ChR2/PV should be done in the PV-ChR2 mice using immunostaining.

We now added images of PV immunostaining in PV-ChR2 mice to Figure 4A and Figure 4—figure supplement 1, as well as co-localization numbers to the text.

2) Please show that the slow GPSCs in adult are blocked by a GABAA receptor antagonist.

We have added new traces and data showing blockade of PV-GPSCs with gabazine in Figure 1 and Figure 6—figure supplement 1.

3) Paired PV-immature GC recordings (currently n=5 with 2 that showed spillover) could offer some additional insights into spillover transmission, namely in terms of whether it is somewhat cell-specific vs. "volume" transmission, etc.

To address the request for additional insight into whether PV spillover signaling is cell-specific versus “volume transmission”, we have now provided new experiments directly comparing volume and spillover transmission to newborn and mature GCs and additional discussion of the comparison. Volume transmission refers signaling from neurogliaform/ivy interneurons that release GABA into the extracellular space rather than at conventional postsynaptic anatomical specializations (Szabadics, 2007; Olah, 2009; Karayannis, 2010). We previously described electrically-evoked and unitary volume transmission to Pomc-EGFP newborn GCs (Markwardt et al., 2009, 2011) and now have confirmed optogenetic-evoked volume transmission from nNos-expressing neurogliaform interneurons (new Figure 3C, D, new Figure 3—figure supplement 1). We consider this to be cell type-specific signaling because it represents the normal mode of transmission for neurogliaform interneurons and thus GPSCs in newborn and mature GCs have similar characteristics indicative of volume transmission. In contrast, PV signaling to newborn and mature GCs has distinct characteristics that result from GABA released at mature GC synapses “spilling over” to act on newborn neurons. We consider this a form of non-specific signaling because spillover GABA is acting beyond the target established by anatomical synaptic connectivity. Discussion added in the subsection “Slow GPSCs from multiple interneuron subtypes”.

We think the direct demonstration of spillover versus volume transmission (Figure 3C, D, new Figure 3—figure supplement 1) is more informative than adding more n’s using paired recordings. The description of paired PV-immature GCs recordings (2/5 successes) might mistakenly give the impression that unitary GPSCs were frequent. However, we previously tested a large number of unlabeled interneurons for connectivity with newborn GCs and found a very low success rate (2.3%; 11 of 498 attempts), with none from PVs (Markwardt et al., 2011). Thus, the goal of experiments in Figure 6 was to illustrate that light-evoked GPSCs arise from multiple PVs as expected for transmitter “pooling” from multiple axons and release sites. We were surprised to find that in 2 recordings (on the same day), we evoked a unitary GPSC by direct stimulation of the PV. We’ve added rise and decay times for those unitary GPSCs to the text. But as other attempts at paired recordings have been unsuccessful, we don’t think this is a useful approach. Rather we conclude that it is possible to generate a spillover GPSC from a single PV, but optogenetic activation results from cooperative GABA pooling from many interneurons. We have altered the text to clarify these points.

4) One surprising result in Figure 2G is that full blockade of the α1 subunit responsible for the fast kinetics produces no change in the rise time in new or mature GCs, in contrast to what would have been expected, an increase in rise time. This inconsistency must be addressed.

While at first glance this might be surprising, our negative results are fully consistent with known determinants of compound or multi-synaptic responses in which additional factors, like asynchrony of release and transmitter pooling, make substantial contribution to the time course of evoked currents. This is why evoked IPSCs (and EPSCs) have slower rise and decay phases than mIPSCs/mEPSCs generated at the same synapses (i.e. Diamond and Jahr, 1995; Overstreet- and Westbrook, 2003). Further, the time course of GPSCs evoked by dendritic-projecting neurogliaform interneurons is dictated primarily by the spatial-temporal profile of [GABA] rather than receptor subunit composition (i.e. Szabadics et al., 2007; Karayanis et al. 2010). We have provided additional explanation of these points and cited some reviews addressing them (subsections “Differential Expression of α1 subunit cannot account for slow GPSCs” and “Mechanisms underlying slow GPSCs from PVs”).

Another way to confirm that factors other than subunit composition contribute to the difference in GPSCs between newborn and mature GCs is to compare the isolated α1 receptor-mediated GPSCs (since young GAD67-GPF cells have low levels of α1 receptors). If receptor kinetics dominant the IPSC kinetics, we would expect that α1 mediated currents will have the same kinetics regardless the transmitter profile. However, we observed that α1 mediated GPSC in young GCs have slower kinetics compared to that in mature GCs (new Figure 2—figure supplement 1). This result also indicates that differences in other subunits expression cannot account for the kinetic differences.

5) IPSC decay was faster after pre-conditioning pulses, presumably due to less GABA release. By this argument, the second IPSC during paired-pulse experiments in which there is paired-pulse depression should also be faster – is this the case?

This is a good point. We now report that the decay of the second IPSC in paired-pulse experiments is indeed faster than the decay of the first IPSC in both mature and newborn GCs. We have added this data to the subsection “PVs generate spillover to mature GCs”.

This should be addressed convincingly, using either new experiments (TPMPA) or with reanalysis of existing data if they are convincing. E.g., one might expect that the second response (in immature neurons) would be more sensitive to TPMPA than the first one.

We agree that fewer vesicles released following the second stimulus will likely generate a lower average [GABA] at receptors on newborn GCs compared to the first stimulus. However, TPMPA is insufficient to detect an expected 2-3-fold difference in [GABA]. To put this in perspective, we found that TPMPA generates a <20% differential block between GPSCs in mature and newborn GCs (Figure 3). Previous work indicates that a 20% differential block by TPMPA corresponds to ~ 100x difference in peak [GABA] between IPSCs at conventional synapses (est. peak [GABA] of 2-3 mM) and spillover-like transients from neurogliaform interneurons (est. peak [GABA] of 20-40 µM; Karayannis et al., 2010). Thus, if a 20% differential block corresponds to a 100x difference in peak [GABA], TPMPA would not be able to differentiate a much smaller difference in the [GABA] mediating the first and second spillover GPSC. Assessing the effect of TPMPA on the PPR is further complicated by evidence that TPMPA reduces frequency-dependent depression of spillover-like transients by protecting postsynaptic GABAA receptors from entering slow desensitized states that enhance frequency-dependent depression (Karayannis et al., 2010). We thank reviewers for reminding us to also make the point that the strong PPD at newborn synapses is consistent with a contribution of postsynaptic desensitization (now included in the subsection “Mechanisms underlying slow GPSCs from PVs”).